# Ultrasound Treatment Enhanced Semidry-Milled Rice Flour Properties and Gluten-Free Rice Bread Quality

**DOI:** 10.3390/molecules27175403

**Published:** 2022-08-24

**Authors:** Wanyu Qin, Huihan Xi, Aixia Wang, Xue Gong, Zhiying Chen, Yue He, Lili Wang, Liya Liu, Fengzhong Wang, Litao Tong

**Affiliations:** Key Laboratory of Agro-Products Processing Ministry of Agriculture, Institute of Food Science and Technology, Chinese Academy of Agricultural Sciences, Beijing 100193, China

**Keywords:** ultrasound, semidry-milled rice flour, rice bread, properties

## Abstract

The structural and functional properties of physical modified rice flour, including ultrasound treated rice flour (US), microwave treated rice flour (MW) and hydrothermal treated rice flour (HT) were investigated with wet-milled rice flour (WF) used as a positive control. The results showed the presence of small dents and pores on the rice flour granules of US and MW while more fragments and cracks were showed in HT. XRD and FTIR revealed that moderate ultrasonic treatment promoted the orderly arrangement of starch while hydrothermal treatment destroyed the crystalline structure of rice flour. In addition, the significant decrease of gelatinization enthalpy and the narrowing gelatinization temperature were observed in US. Compared to that of SF, adding physical modified rice flour led to a batter with higher viscoelasticity and lower tan δ. However, the batter added HT exhibited highest G′ and G″ values and lowest tan δ, which led to a harder texture of bread. Texture analysis demonstrated that physical modified rice flour (except HT) reduced the hardness, cohesion, and gumminess of rice bread. Especially, the specific volume of bread with US increased by 15.6% and the hardness decreased by 17.6%. This study suggested that ultrasound treatment of rice flour could improve texture properties and appearance of rice bread.

## 1. Introduction

Due to the increase in the prevalence of gluten-related diseases, such as celiac disease, gluten-free bread has a growing demand and consumption [1]. Rice flour is the major ingredient frequently used in the manufacture of commercially available gluten-free bread since it is colorless, bland taste, easy digestibility and hypoallergenic [2]. However, rice flour lacks gluten to impart viscoelasticity to the dough, which limited the industrial range of native rice flour. Therefore, some modification methods, including mechanical, physical, chemical and enzymatic methods, are usually used to improve the physicochemical properties of rice flour [3,4,5]. Physical modification is the preferred modification method for environmental and economic considerations, which can change the properties of water absorption, gelatinization and thermal properties by affecting the molecular structure of starch or protein, so as to achieve satisfactory processing performance.

Studies have shown that hydrothermal treatment, microwave and ultrasound are commonly used as physical modifications. Hydrothermal treatment is based on treating materials with moisture content less than 35% (*w*/*w*) within a certain temperature range, which would destroy the crystallization structure of starch and change the solubility, viscosity and expansion ability of rice flour. Ruiz, Srikaeo and Revilla [6] found that the hydrothermal treatment reduced the solubility, expansion capacity and peak viscosity of rice starch, and increased its gelatinization temperature. Microwave technology is based on the dielectric heating effect, leading to the rapid dipole reorientation of polar molecules in the starch system, which changes the structure and properties of starch. Li et al. [7] observed that pores and cracks appeared in the starch particles with 30% water content under microwave treatment. Villanueva et al. [8] adopted microwave-assisted hydrothermal treatment to modify rice flour, which found that the pasting temperature was delayed and the pasting viscosity was reduced, and the deformation resistance and recovery ability of the dough were improved. Moreover, ultrasound wave could cause structural changes to the starch by producing cavitation effect and generating free radicals [9,10]. Cui and Zhu [11] discovered that ultrasound treatment changed the morphology of starch particles and decreased its crystallinity, and the gelation enthalpy and pasting viscosity of sweet potato powder were significantly reduced with increasing treatment time. To our knowledge, level 6 structure of starch refers to the whole grain structure of starch particles after interaction with other biomolecules [12]. Although most studies have focused on starch particles after modified treatment, the change of the complex whole grain structure during processing is the main determinant of food quality. Thus, how these three physical modification methods may change the physicochemical properties of rice flour and further affect the rice bread quality remains to be better studied.

In this research, three physical modified methods, including hydrothermal treatment, microwave and ultrasound, were applied on semidry-milled rice flour. The effects of physical modification on the structural characteristics, including morphology, particle size, crystal structure and short-range ordered structure, were analyzed. Changes in physicochemical properties of modified rice flour were investigated for hydration properties, thermal and pasting properties, and rheological properties. The quality of gluten-free bread added physically modified rice flour were evaluated. As one of the common commercial rice flour raw materials, wet-milled rice flour was employed as a reference for comparison. It is expected that this research may provide new perspectives for the application of physically modified flour in gluten-free rice bread.

## 2. Results and Discussion

### 2.1. Particle Size, Damaged Starch (DS)

According to Qin et al. [13], rice flour with particle size <100 μm (D50) showed more suitable properties for bread making. The D50 of all the samples was lower than 100 μm in Table 1. The particle size distribution of US was more uniform in Figure 1, and its average particle size was almost 2.47 times lower than that of SF, which may be conducive to the uniform distribution of starch granules. Cui et al. [11] found that with the increase of ultrasonic power, the particle size of sweet potato and wheat flours gradually decreased, which was consistent with our study.

The DS content determined in rice flour was also shown in Table 1. The relatively lower content of DS in WF (2.06 g/100 g) and SF (3.96 g/100 g) was due to the soaking and tempering process that could soften rice grains, reduce the input of mechanical force and obtain more complete starch granules [14]. Meanwhile, MW had similar contents of DS (3.33 g/100 g), which was associated with the low output power of the microwave process. However, ultrasonication and hydrothermal treatment create significant damage to the integrity of flour particles, the corresponding DS content was 9.91 g/100 g and 24.41 g/100 g, respectively. The observed results are consistent with the smaller particle size of US and HT. During ultrasonic treatment, the granules of cereal starch easily disrupted by the generation and rapid disintegration of gas bubbles in the cavitation effect, and eventually lead to damaged particle [15]. As for HT, the treatment temperature was higher than the gelatinization temperature, the starch underwent an irreversible transition and lost crystallinity. Freeze-drying followed by grinding further led to an increase in DS content.

### 2.2. Granule Morphology

The micrographs of SEM analysis are presented in Figure 2. WF and SF granules were polyhedral in form and regular in shape with a smooth surface. Rice starch were tightly wrapped in rice granule cells and closely linked to protein bodies and lipids [16]. Additionally, the microwave treated rice flour maintained the same particle integrity as the control flour (WF, SF). The subsequent magnification revealed that there were some dents and roughness on the surface of MW (white arrow in Figure 2B-MW). After ultrasonication treatment, the surface depression, pores and channels of rice flour became more obvious, which go in line with the results of DS (Table 1). The cavitation effect and mechanical effect of ultrasonication resulted in slight disruption on the surface of rice starch, which was in agreement with the observation of Kaur and Gill [15].

The modified granular structure using hydrothermal as compared to the other rice flour exhibited significant variations in shape and degree of agglomeration. These granules appeared to have a honeycomb-like structure (Figure 2A-HT). More fragments and cracks appeared on the surface shape, and the starch particles were swollen and gelatinized, resulting in particle aggregation (Figure 2B-HT). The gelatinization of starch was due to the higher temperature of the hydrothermal process. The microstructure changes of rice flour may affect its physicochemical properties, such as water absorption capacity and so on.

### 2.3. Crystal Structure

The X-ray diffraction patterns of native and physical modified rice flour are presented in Figure 3. Physically modified rice flour showed similar X-ray diffraction pattern to the control groups except HT, and they displayed the typical A-type crystal characteristics with strong diffraction peaks around 15°, 17°, 18° and 23° at 2θ. However, the crystalline structure of HT was completely damaged, showing a non-crystalline state [17]. The relative crystallinity calculated from the X-ray diffractogram pattern slightly decreased in US and MW, whereas it significantly decreased in HT. Yang et al. [18] reported that the non-crystalline regions of starch granules were more easily damaged than the crystalline regions. Therefore, we speculated that ultrasound mainly destroyed the non-crystalline regions of US in this study. In addition, the high-density energy input makes the water molecules in the starch particles boil and evaporate quickly without moving out, and high internal vapor pressure was generated during microwave treatment. The existence of pressure and the rapid migration of water may cause the decrease in MW crystallinity [19].

### 2.4. Short-Range Ordered Structure of Starch and Secondary Structure of Protein

Besides investigating the long-range order of the double helices in rice starch by XRD, the short-range order structure was also determined by FT-IR. The absorbance ratio of 1047/1022 cm^−1^ has been widely applied to describe the degree of short-range order in starch. The bands at 1047 and 1022 cm^−1^ represented the crystalline order and the amorphous region of starch, respectively. As shown in Table 1, compared to SF, the R1047/1022 value of MW remained unchanged, but increased in HT and US. This indicated that appropriate ultrasonic irradiation enhanced the recrystallization of starch and improved the ordered crystalline region. However, the study of Vela, Villanueva and Ronda [20] reported that the R1047/1022 value of rice starch was reduced after ultrasound treatment, owing to the breaking of hydrogen bonds. The inconsistent result may be due to the differences in varieties and experimental conditions, such as flour concentration, treatment time and treatment temperature. Besides, the R1047/1022 value of MW was slightly lower than that of SF, which might be attributed to the rapid boiling and evaporation of water caused by microwave heating. This was similar to the result of Han et al. [21].

The absorption peak of -OH (3200*–*3500 cm*^−^*^1^) represents the strengths of the intermolecular hydrogen bonds in rice flour, and is presented in Figure 4. The shift of the peak to the low frequency wave number (red shift) implies an increase in the strength of intermolecular hydrogen bonds, while the shift to the high frequency wave number (blue shift) indicates a decrease in the strength of hydrogen bonds [22]. Figure 4 showed that the -OH absorption peaks in HT and MW blue shifted to the high frequency wave numbers, 3418.43 and 3406.44, respectively. This result may be due to the fast vibration of polar molecules caused by microwave heating and the disintegration of the double helix structure after gelatinization caused by hydrothermal treatment. While the -OH absorption peaks in US red shifted to the high frequency wave number compared to that of SF, indicating that ultrasound strengthened the intermolecular hydrogen bond in rice flour. With the increase in ultrasonic intensity and time, the loosened proteins reassembled again to form new hydrogen bonds, and the starch short chains were further linked by hydrogen bonds to form a new ordered structure [23].

In order to explore the effect of physical modification on the protein secondary structure, the change in amide I (1700*–*1600 cm^−1^) was estimated from the relative area of each individual peak shown in Table 1. Hydrothermal treatment led to a significant reduction in α-helix up to 45.1% and an increase in random coil up to 34.3% in HT compared to SF. This result indicated that hydrothermal treatment could cause the unfolding and rearrangement of polymeric subunits, and eventually lead to disordered structure. There was an increase in α-helix content of US, but the β-turn content of US decreased. These observations may be explained in part by the reorganization of the loose protein structure under cavitation effect of ultrasound. These findings were inconsistent with Vela et al. [24], who reported that ultrasound decreased the content of α-helix and β-sheet in rice protein. This differing result might be related to the differences in protein and sonication conditions. In MW, the increase of β-sheet was accompanied by a decrease in β-turn, indicating the transformation between these two structures, and microwave treatment helped implement the conversion from β-turn to β-sheet. This change may improve the texture of rice bread. According to a previous study in wheat flour, the rise of β-sheet and the decline in β-turn made the gluten network of dough more ordered, which reduced the hardness of steamed bread and improved the quality [25].

### 2.5. Hydration Properties

Water absorption index (WAI), water solubility (WS) and swelling power (SP) of rice flour with different physical modifications are shown in Table 2. The results exhibited that hydration properties values of SF and WF in 25 °C were statistically insignificant (*p* > 0.05) and lower than (*p* < 0.05) that of modified rice flour. In contrast, WAI, WS and SP of HT was highest in all samples. The higher hydration properties values could be due to the highest DS content and the cracks and fragments of the damaged granules, which made the granules easier to absorb water and expand [26]. In addition, high values of SP in HT could be caused by the damage of the crystalline structure of rice starch (Figure 3) and the combination of water molecules with the free hydroxyl groups in amylose or amylopectin [27]. Ultrasound treatment also significantly increased the WAI and WS of rice flour in 25 °C, which may be because ultrasonic treatment destroyed the amorphous region of starch particles, resulting in amylose leaching and increased solubility.

### 2.6. Thermal Properties

The influence of physical modifications on rice flour thermal properties was determined and the results are shown in Table 2. Higher To, Tp, Te and ∆H were observed in WF compared to SF, which was due to the complete starch granules and fine flour particles, leading to a greater barrier for energy transfer and higher heat stability [13]. To, Tp, Te and ∆H were not detected in HT since the starch granules was completely gelatinized after hydrothermal treatment. Compared with SF, no significant (*p* > 0.05) changes were observed in the value of gelatinization temperature and ΔH of MW. Nevertheless, the ΔH of US presented a decreased trend, which might be related to the higher DS content and the weakening of amorphous regions. The pores and cracks on the surface of DS (Figure 2), as well as the destruction of amorphous areas caused by cavitation effect (Section 3.3), make it easier for water to diffuse into particles and further penetrate into the crystalline area of starch. Decreased ΔH was also reported for rice starch after ultrasonication [18]. Meanwhile, the end temperatures were significantly decreased after ultrasonication (74.33 °C) compared to that of SF (79.32 °C), leading to a narrowed ∆T. This result is mainly due to the reinforcement of starch structure induced by ultrasound treatment. The ultrasonic treatment distorted amorphous regions of the starch granules (Figure 3), increased the starch homogeneity and thus strengthened the remaining crystalline structure (Table 2). As a result, the energy required for gelatinization (∆H) was reduced and the gelatinization temperature (∆T) was narrowed. The results are consistent with the reports of Amini, Razavi and Mortazavi [28] and Chi et al. [29].

### 2.7. Pasting Properties

Pasting properties of rice flour with different physical modifications are shown in Table 2. MW exhibited much higher pasting viscosity compared to those of other physical modified rice flours and close to those of WF. HT exhibited lower values of pasting viscosity than those of SF, as a consequence of the molecular destruction and gelatinization of starch granules during hydrothermal treatment. In addition, sonication led to a significant decrease in the viscosity profile, consistent with the study in corn starch by Yang et al. [30]. Cracks on the surface of US starch particles contribute to the penetration of water for hydration, causing significant decrease in viscosity. Besides, SB value of starch was related to the amylose structure, which mainly constituted the amorphous regions. Ultrasound treatment distorted the amorphous regions, resulting in the leaching out of amylose molecules and a decrease in SB. However, an opposite result was observed by Yang et al. [18], who found an increase in pasting viscosity in ultrasound treated rice starch compared to native starch. It could be explained by the secondary structure of rice protein. Ultrasound treatment may reduce proteins folding, which could increase the sites for binding water. This would reduce the possibility of other components contacting water and decrease the rate and extent of leaching out of granular components from the granule, thereby reducing the pasting viscosities [31].

### 2.8. Rheological Properties of Rice Batter

The viscosity and elasticity of batter are the key indicators of bread quality [32]. The mixed rice flour could form a starch-based batter structure. The function of HPMC is to simulate the structure of gluten, to improve the viscoelasticity of the paste and stabilize the gas in the mixing process [33]. The dynamic viscoelastic properties of batter added different modified rice flour were investigated in Figure 5. Frequency sweeps showed that G′ exceeded G″ in the whole frequency range tested and tan δ < 1 at any point, indicating the solid-like behavior of all the batter. With the increasing frequency, G′ and G″ were increased, showing that viscoelasticity was enhanced. The G′ and G″ values of the modified rice flour were higher than those of SF, and the tan δ curve of the modified rice flour was lower than that of SF. These results indicated the viscoelasticity of the batter was enhanced and the gel strength of the batter was improved after adding physical modified rice flour. The rheological properties of the batter with US were the closest to those of WF, suggesting that batter with US had more viscoelastic and structured. This was probably because the amorphous regions of starch were destroyed by sonication, which promoted the leaching of amylose and the extension of the helix structure increased the association of linear chains, which was able to form a consolidated network [34]. The viscoelasticity of the batter added MW also improved compared to that of SF. However, batter added HT exhibited highest G′ and G″ values and lowest tan δ, which indicated a harder texture. This was because HT with a high DS content was easier to absorb water, leading to excessive intermolecular cross-linking of the batter, which would negatively affect the development of the dough network and lead to a stiffer bread crumb [35].

### 2.9. Bread Quality

Table 3 and Figure 6 show the specific volume and microstructure of rice bread added physically modified rice flour. It can be found that the specific volume of rice bread with the addition of US (3.63 mL/g) and MW (3.51 mL/g) was greater than that of SF (3.14 mL/g), and comparable to that of WF (3.72 mL/g). In order to obtain a large volume of bread, the dough not only need to be soft enough to expand during proofing, but also have enough strength to bear the rapid expansion of CO_2_ gas cell during baking. The higher bread volume could be related to the increase of batter viscosity as a result of ultrasound and microwave treatment. The rheological properties (Figure 5) showed that the viscoelasticity of batter with US and MW were improved compared to SF, and the elasticity and viscous modulus of the batter with US were closest to those of WF. The increase of dough viscoelasticity improved the gas retention ability of dough, and prevented gas from escaping during fermentation and baking, which could produce rice bread with higher specific volume to a certain extent [36]. Therefore, the bread with US and MW had higher volume compared to the SF formulation. However, excessive viscosity would reduce the ductility of the dough and limit the generation and maintenance of gas, resulting in a low specific volume of bread [37]. This would explain why the batter with 15% of HT, having highest elasticity and viscous modulus and lowest tan δ, led to breads with the smallest bread volume (1.18 mL/g).

From the microstructure of breadcrumbs shown in Figure 6B, it can be seen that the network structure formed by gelatinized starch, HPMC and denatured protein showed a smooth surface, dense and porous, and was similar to the honeycomb gluten network. Under the microscope, the gas cells of the bread added physically modified rice flour became larger. However, the cohesion of the network structure of the bread with HT was enhanced and the matrix tended to thicken. This occurred because the excessive cross-linking of the batter with HT resulted in a compact structure, which limited the expansion of the bread and was not conducive to the edible quality of the bread.

Texture properties of the rice bread added physically modified rice flour are shown in Table 3. Hardness is usually considered a key indicator of sensory perception, which is related to the specific volume of the rice bread. A higher specific volume means a higher amount of air retained in the crumb structure, leading to a softer texture [38]. The use of US and MW make breadcrumbs softer compared to SF, corresponding to a 17% and 10% decrease in hardness, respectively. Studies have shown that the cavitation of ultrasound can destroy the amorphous region of starch, leading to the exposition of hydrophilic groups and the increase of dough hydrophilicity, which reduce the hardness [25]. With the increase of ultrasonic time, the short-chain molecules rearranged to produce a more compact structure, and the linear molecules were more tightly bound to form a new ordered structure, which facilitated the generation and maintenance of the network structure of the dough and bread. Whereas, the microwave treatment destroyed the hydrogen bonds within the molecule and the protein secondary structure (Section 3.4), broke the long chains of starch into short chains, which may contribute to the soft texture of rice bread. The same result was previously reported by Villanueva et al. [8], who found that the partial replacement of rice flour by microwave treatment significantly increased bread specific volume. However, the addition of HT led to a harder crumb (from 51.30 g to 701.70 g). The small bread volume corresponded to less amount of air retained in the batter during proofing and baking, mainly due to the excessive dough viscoelasticity, which endorsed a stiffer crumb. Similar results were observed in gumminess and chewiness, probably because these indicators were mainly affected by hardness. However, the resilience, cohesion and springiness of the bread added different physical modified rice flour were overall lower than that of the control bread.

### 2.10. Principal Component Analysis (PCA)

Figure 7 shows the results of principal component analysis for rice flour and rice bread attributes, to explore the relationship of rice flour with different physical modifications and rice bread added physical modified flour. Overall, PC1 and PC2 accounted for 80.28% of the total variation, indicating the plane of PC1 and PC2 reflects, to a large extent, the main contributions of the response variables. The results show that different physical modifications led to significantly different flour properties and bread quality. In Figure 7, according to the properties of rice flour and rice bread, the samples are divided into three groups as a whole. HT was located in the positive quadrant of PC1 and PC2, which was separated from the rest of the samples since it possessed high DS content, water solubility and water absorption, and the bread with HT had a stiffer crumb texture. This result indicated that hydrothermal treatment could destroy the rice starch, increase the DS content and make the granules absorb water easier, which lead to the hard crumb texture and high chewiness of bread. SF and MW were located in the negative part of PC1 and PC2. US and WF were located in the negative quadrant of PC1 and the positive quadrant of PC2, and their bread exhibited high specific volume as well as high springiness, cohesion and resilience, which was due to the high extensibility and resistance of batter adding US and WF. In general, adding ultrasonic modified rice flour would improve the quality of bread based on semidry-milled rice flour, whereas hydrothermal treatment is not a desirable modification method.

## 3. Materials and Methods

### 3.1. Rice Flour

Polished japonica rice (Panjin, China; harvested in 2020) provided by Panjin Pengyue Rice Industry Co., Ltd. was used in this study. The moisture content was 11.46%, protein: 8.21% dry basis (db), total starch: 80.75% db and amylose: 18.80% db, respectively. Semidry-milled rice flour (SF) was prepared according to Qin et al. [13]. A total of 200 g of polished japonica rice was first treated by hot air in an oven at 45 °C for 1 h and subsequently tempered for 35 min with the addition of water. The added water volume was calculated as follows:Added water (mL) = m × (ω1 − ω2)*/*(1 − ω1)(1)
where m represented the weight of rice flour, ω1 represented the saturated moisture content of rice grain (approximately 28% according to preliminary experiment) and ω2 represented the moisture content of rice grain after hot air treatment. After grinding the rice with a cyclone mill (CT410, Foss Analytical Co., Ltd., Hillerød, Denmark), the obtained rice flour was dried in an oven at 45 °C until the moisture content was less than 12%.

Wet-milled rice flour (WF) was prepared as follow: 200 g of polished rice grains were soaked in 400 mL water for 2 h and then milled in a colloid mill (JMS-30A, Langtong Machinery Co., Ltd., Hebei, China). The obtained slurry was freeze-dried (SCIENTZ-10N, Scientz Biotechnology Co., Ltd., Ningbo, China) and then sifted through a 100-mesh screen. In this study, SF and WF were set as the control.

### 3.2. Different Physical Modification Methods

#### 3.2.1. Ultrasound Treatment

SF was suspended in distilled water to give a concentration of 30% (*w*/*v*) and stirred for 10 min to obtain a homogeneous slurry. An ultrasonic processer (ScientzIID, Zhejiang, China) equipped with a 6 mm titanium alloy probe was used. According to the method of Yang et al. (2019) [18] with minor modifications, the slurry was treated with ultrasound for 20 min (5 s on and 5 s off) at a constant frequency of 20 kHz with an output power of 600 W, and the probe was placed at 1.5 cm depth. Then, they were retrieved by freeze-drying, followed by sifted through a 100-mesh screen. SF treated by ultrasound was coded as US and stored in a desiccator at room temperature.

#### 3.2.2. Microwave Treatment

A 100 g of SF was put in a glass container and then placed in the center of the microwave oven (M3-L205C, Midea, Foshan, China). As per the previous method with slight modifications [19], samples were irradiated at 300 W for 5 min. The irradiated samples were freeze dried and then sifted through a 100-mesh screen. SF treated by microwave was coded as MW and stored in a desiccator at room temperature.

#### 3.2.3. Hydrothermal Treatment

Hydrothermally treated rice flour (HT) was prepared by referring to the method of Bourekoua et al. [39] with minor modification. SF was suspended in distilled water to obtain 20% (*w*/*w*) suspension and stirred continuously in a hot water bath until the inner temperature reached 65 °C. The rice slurry was heated in a hot water bath for 30 min. Then, the paste was cooled at room temperature and freeze-dried. The dried rice was milled into flour by a grinder and passed through a 100-mesh screen. The sample was stored in a desiccator at room temperature.

### 3.3. Particle Size Distribution, Damaged Starch (DS) Content and Protein Content

The particle size of all samples was determined by Malvern MasterSizer 3000 (Malvern Instrument, Ltd., Worcestershire, UK). In addition, D50 (the particle diameters at cumulative volume percentage of 50%) was calculated by the software provided by the device. The DS content (AACC Method 76–31.01) was measured by commercial assay kit (K-SDAM, Megazyme International Ltd., Wicklow, Ireland).

### 3.4. Scanning Electron Microscopy (SEM)

Rice flour was fixed onto the surface of a circular aluminum stud with double-sided sticky tape and then observed using a scanning electron microscopy (Hitachi S-570, Hitachi, Co., Ltd., Tokyo, Japan) under 10 kV.

### 3.5. X-ray Diffraction (XRD)

The crystallization characteristics of freeze-dried rice flour were determined by X-ray powder diffractometer (Bruker D8 Advance, Salbuluken, Germany). Scans were performed at diffraction angles (2θ) ranging from 5° to 45°, with a scanning rate of 10°/min and step size of 0.02°. Then, 40 kV and 40 mA with Cu-Kα radiation were set as operating conditions. The relative crystallinity (%) of all samples was calculated by MDI Jade 6.0 software (Materials Data, Inc., Livermore, CA, USA).

### 3.6. Water Hydration Property

Water hydration property included water absorption index (WAI), water solubility (WS) and swelling power (SP). Additionally, 100 mg of rice flour was dispersed in 10 mL distilled water and stirred for 30 min at 25 °C. The suspensions were then cooled to room temperature and centrifuged at 8500× *g* for 30 min. Subsequently, the sediment was weighed for calculating WAI. The supernatants were collected in an aluminum dish of a known weight, dried at 105 °C until the weight was constant [40]. WAI, WS and SP were calculated according to the previous study [13].

### 3.7. Short-Range Ordered Structure of Starch and Secondary Structure of Protein

The FTIR spectra of rice flour samples were obtained using the FTIR spectrometer (TENSOR 27, Borken, Germany). Samples were equilibrated at 45 °C for 24 h and mixed with KBr (1:75, *w*/*w*), followed by grounding and pressing into tablets. The wavenumber range was set in the range of 4000–600 cm^−1^ at a resolution of 4 cm*^−^*^1^ with 64 scans. The spectrum was normalized by using OMNIC (Thermo Fisher Scientific, Waltham, MA, USA). The changes in the short-range order structure of starch were calculated by the absorbance ratio 1047/1022 (R1047/1022). Amide I bands (1700*–*1600 cm*^−^*^1^) were analyzed using PeakFit 4.12 (SeaSolve Software Inc., San Jose, CA, USA).

### 3.8. Differential Scanning Calorimetry (DSC)

The thermal characteristics of physical modified rice flour were measured by a DSC8000 analyzer (PerkinElmer, Norwalk, CT, USA). Then, 3 mg of rice flour was accurately weighed in an aluminum pan, and deionized water was added at a ratio of 1:2 (*w*/*w*). The sealed samples were balanced overnight at 4 °C and scanned at a temperature range of 30*–*110 °C at a rate of 10 °C/min. A sealed empty pan was used as a reference [41].

### 3.9. Pasting Properties

A Rapid Viscosity Analyzer (RVA-TecMaster, Perten Instruments, NSW, Australia) was adopted to measure the pasting properties of the samples. Each sample (3 g, dry basis) was mixed with 25 g distilled water before being transferred into an aluminum canister. The sample was equilibrated at 50 °C for 1 min, heated from 50 °C to 95 °C at 0.2 °C/s, held for 2.5 min at 95 °C, then cooled to 50 °C at the rate of 0.2 °C/s, and finally maintained at 50 °C for 2 min [13].

### 3.10. Rheological Properties of Gels

The dynamic viscoelasticity of rice batter (consisting of all ingredients except butter and yeast) were assessed by a Rheometer (Physica MCR 301, Anton Paar GmbH, Graz, Austria). The resting time of batter samples before the measurement was 5 min. Dynamic viscoelasticity measurements were done using parallel plate geometry (50 mm diameter, 1 mm gap) at 25 °C. Conduct frequency scanning of 0.1*–*100 rad/s with 0.01% strain [42]. Three replicates were performed for each analysis.

### 3.11. Breadmaking Procedure

The physical modified rice flour of US, MW and HT were prepared into bread, and mixed with SF in the ratio of 3:17 (For example, 15 g modified rice flour and 85 g SF). The gluten-free rice bread formula was based on 100 g mixed rice flour, 90 g water, 4 g HPMC, 20 g sugar, 10 g batter and 2 g yeast. Yeast was first dissolved in water at 28*–*30 °C to hydrate. The other dry ingredients were mixed evenly for 5 min using a Pin Mixer (JHMZ200, Beijing Dongfu Jiuheng Instrument Technology Co., Ltd., Beijing, China). Then, the hydrated yeast mixture and batter were added and mixed for 5 min. After stirring, 50 g of batter was filled in a baking tray (70 mm × 70 mm × 40 mm) and proofed in a fermentation chamber (CF-6000, Zhongshan Kashi Electric Appliance Co., Ltd., Zhongshan, China) for 40 min at 38 °C and 85% relative humidity. The batter was put in an electric oven (T7-L328E, Midea Group Co. Ltd., Foshan, China) and the baking parameters were set at 150 °C for 15 min. After baking, the bread was taken out and then cooled for 1 h. Finally, the samples were sealed and stored for determination. Bread was made three times for each formulation.

### 3.12. Gluten-Free Rice Bread Quality

Bread volume (mL) was measured after 1 h cooling and determined by the rapeseed displacement method 10-05 (AACC International, Saint Paul, MN, USA, 2000). The specific volume (mL/g) was calculated as the volume divided by the weight measured 1 h after baking.

After baking and cooling, the breadcrumbs were immediately frozen by liquid nitrogen and then freeze-dried. The size of 4 mm × 2 mm × 2 mm small pieces were cut from the center of the breadcrumbs, and then their microstructures were examined by SEM at magnifications 1000× *g* with 10 kV.

Crumb texture was determined by using a Texture Analyzer (TA-XT2i, Stable Micrio System, Godalming, England). Small piece sizes of 15 mm × 15 mm × 15 mm were cut from the center of the breadcrumbs. The texture profile analysis (TPA) double compression test was performed by an aluminum probe (36 mm diameter; P/36R), which could penetrate 50% depth at 1 mm/s speed test with a 5 s delay between the first and second compression. The trigger force was 5 g. A total of 12 replicates were carried out.

## 4. Conclusions

Rice flour was subjected to ultrasound, microwave and hydrothermal treatments. These physical modified treatments significantly changed the particle size and structure of rice flour, which affected their hydration, pasting and rheological behavior, and further changed the rice bread quality. After hydrothermal treatment, the structure of starch granules was seriously destroyed, and the percentage of irregular curl in the protein secondary structure increased. The excessive batter viscoelasticity of HT limited the development of the batter during fermentation, resulting in a stiffer bread crumb and smaller bread volume. In addition, microwave treatment weakened the hydrogen bonding force and changed the secondary structure of the protein with the conversion from β-turn to β-sheet. The doughs made with MW showed higher viscoelasticity and more stable cell structure than that of SF. Especially, the starch structure of US was reinforced and the protein secondary structure was modified with the decrease in β-turn and increase in α-helix. These molecular changes could explain the decrease of the pasting profile and narrowing of the gelatinization range in US. Moreover, the viscoelasticity of the batter containing 15% US was significantly improved compared to the SF batter, leading to a higher bread volume and softer crumb texture close to those of bread with WF. In conclusion, ultrasound modification is an alternative to improve the morphology and physical properties of rice flour and further improve gluten-free bread quality.

## Figures and Tables

**Figure 1 molecules-27-05403-f001:**
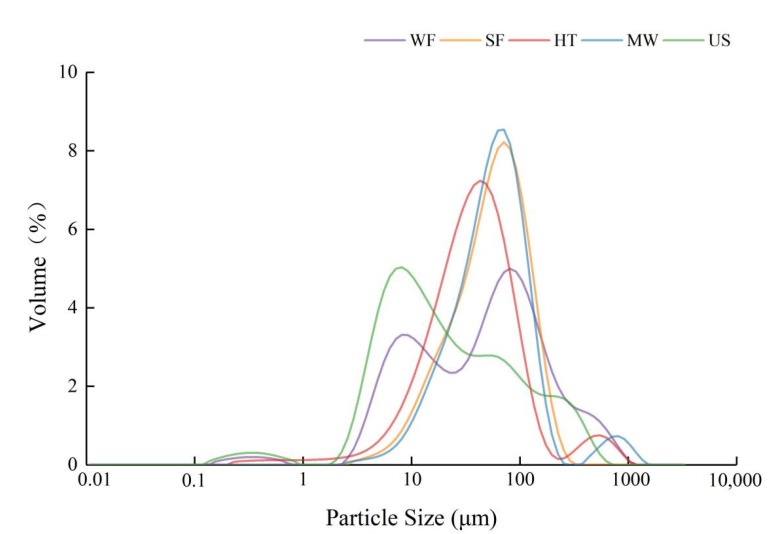
Particle size distribution of rice flour with different physical modifications.

**Figure 2 molecules-27-05403-f002:**
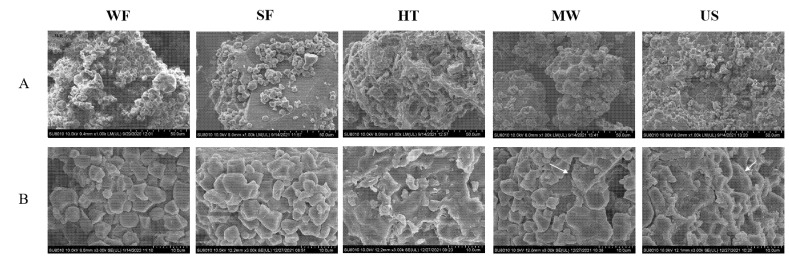
SEM images of rice flour with different physical modifications at magnification of (**A**) 1000×, (**B**) 3000×, respectively.

**Figure 3 molecules-27-05403-f003:**
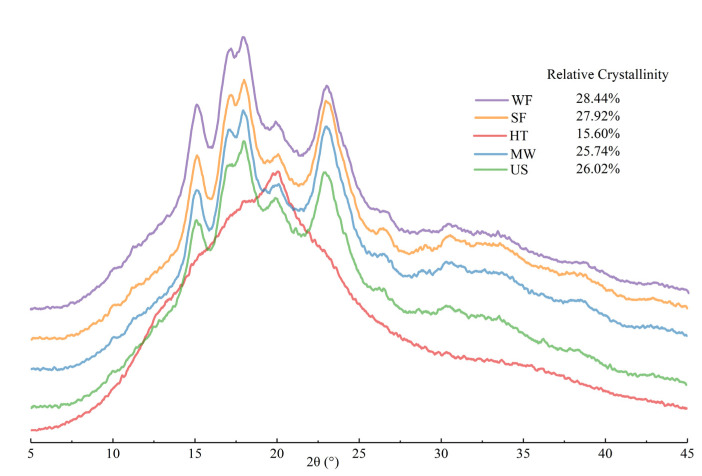
X-ray diffraction spectrum of rice flour with different physical modifications.

**Figure 4 molecules-27-05403-f004:**
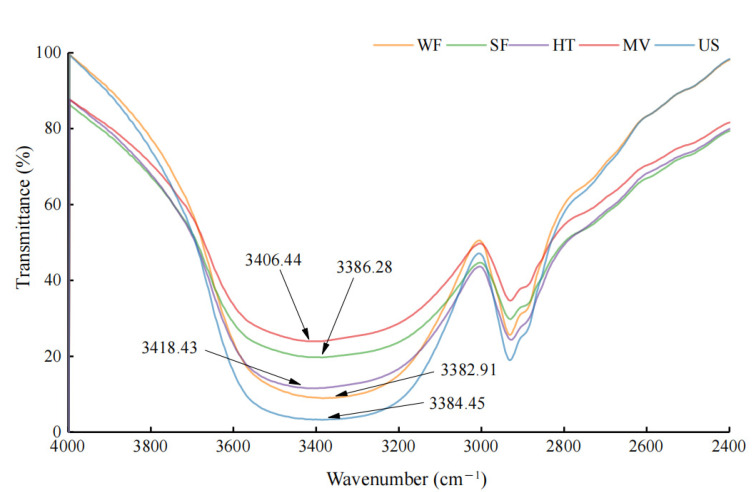
FTIR spectra of rice flour with different physical modifications.

**Figure 5 molecules-27-05403-f005:**
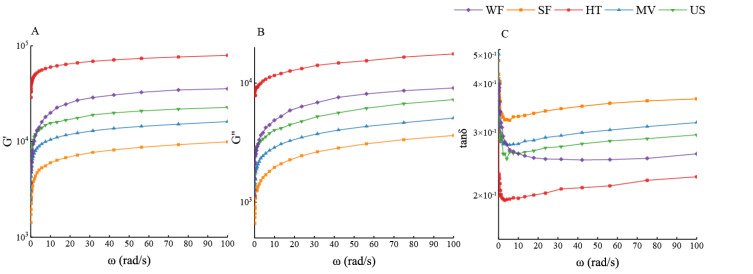
(**A**) The storage modulus (G′), (**B**) loss modulus (G″) and (**C**) the loss tangent (tan δ) of rice flour batter added physically modified rice flour.

**Figure 6 molecules-27-05403-f006:**
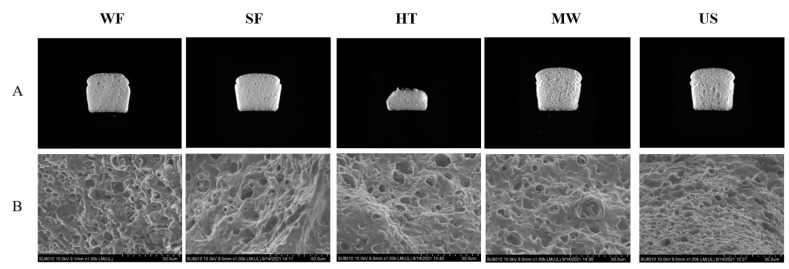
(**A**) Specific volume and (**B**) microstructure of rice bread added physically modified rice flour.

**Figure 7 molecules-27-05403-f007:**
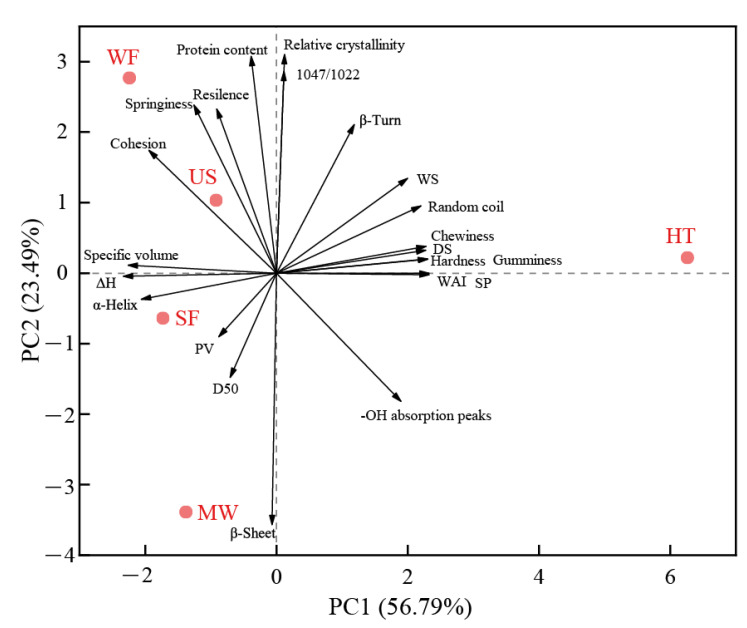
Principal component analysis.

**Table 1 molecules-27-05403-t001:** Particle size, damaged starch, short-range ordered degree (R1047/1022) and protein secondary structure of rice flour with different physical modifications.

Sample	WF	SF	HT	MW	US
D50 (µm)	50.7 ± 0.8c	59.4 ± 0.2a	37.7 ± 1.1d	57.0 ± 1.2b	17.1 ± 0.9e
Damage starch content (g/100 g)	2.06 ± 0.12d	3.96 ± 0.04c	24.41 ± 1.60a	3.33 ± 0.07cd	9.91 ± 0.20b
R_1047/1022_	1.34 ± 0.15ab	1.18 ± 0.05cd	1.28 ± 0.10bc	1.14 ± 0.03d	1.44 ± 0.23a
β-Sheet (%)	26.8 ± 0.7c	30.8 ± 0.2b	30.0 ± 0.5b	35.0 ± 0.3a	30.1 ± 0.3b
α-Helix (%)	25.6 ± 0.5c	27.7 ± 0.4b	15.2 ± 0.5d	28.1 ± 0.3b	31.7 ± 0.7a
β-Turn (%)	27.1 ± 0.4a	21.0 ± 0.6b	27.4 ± 0.5a	18.6 ± 0.2c	18.3 ± 0.7c
Random coil (%)	20.5 ± 0.4b	20.4 ± 0.6b	27.4 ± 0.4a	18.3 ± 0.2c	19.9 ± 0.7bc

Different letters in the same row indicate significant differences (*p* < 0.05). WF: wet-milled rice flour, SF: semidry-milled rice flour, HT: semidry-milled rice flour treated by hydrothermal, US: semidry-milled rice flour treated by ultrasound, MW: semidry-milled rice flour treated by microwave. D50 reflect the particle size by volume at 50% of all particles.

**Table 2 molecules-27-05403-t002:** Hydration, thermal and pasting properties of rice flour with different physical modifications.

Sample	WF	SF	HT	MW	US
Hydration properties					
WAI (g/g)	3.05 ± 0.20c	3.00 ± 0.10c	10.54 ± 0.08a	3.81 ± 0.12b	3.74 ± 0.44b
WS (%)	2.45 ± 0.49b	2.14 ± 0.76b	4.33 ± 0.44a	1.93 ± 0.29b	3.42 ± 0.22a
SP (g/g)	3.13 ± 0.21c	3.07 ± 0.07c	11.02 ± 0.03a	3.89 ± 0.12b	3.88 ± 0.46b
Thermal properties					
To (°C)	62.47 ± 0.11a	60.78 ± 0.45b	-	60.18 ± 0.06b	62.10 ± 0.74a
Tp (°C)	69.02 ± 0.03a	67.36 ± 0.17b	-	66.76 ± 0.17b	66.75 ± 0.75b
Te (°C)	79.98 ± 0.00a	79.32 ± 0.40bc	-	76.26 ± 2.74cd	74.33 ± 0.43d
∆H (J·g^−1^)	8.48 ± 0.02a	7.63 ± 0.19bc	-	7.85 ± 0.17b	7.33 ± 0.30c
∆T (°C)	17.51 ± 0.11a	18.54 ± 0.36a	-	16.09 ± 2.77a	12.22 ± 0.56b
Pasting properties					
PV (cP)	4144 ± 23a	4209 ± 46a	3276 ± 78b	4067 ± 113a	2498 ± 122c
TV (cP)	2580 ± 8a	2300 ± 54b	2449 ± 60a	2506 ± 10a	1731 ± 86c
BD (cP)	1564 ± 15b	1909 ± 13a	829 ± 39c	1561 ± 103b	768 ± 36c
FV (cP)	4218 ± 7a	3853 ± 52b	3901 ± 90b	4182 ± 52a	3099 ± 95c
SB (cP)	1638 ± 1ab	1553 ± 7bc	1452 ± 81cd	1677 ± 43a	1369 ± 29e

WF: wet-milled rice flour, SF: semidry-milled rice flour, HT: semidry-milled rice flour treated by hydrothermal, US: semidry-milled rice flour treated by ultrasound, MW: semidry-milled rice flour treated by microwave. WAI: Water absorption index, WS: water solubility, SP: swelling power, To: onset temperatures, Tp: peak temperatures, Te: end temperatures, ∆H: enthalpy change, ∆T: gelatinization temperature range; PV: peak viscosity, TV: trough viscosity, BD: breakdown viscosity, FV: final viscosity, SB: setback viscosity. Different letters in the same row indicate significant differences (*p* < 0.05).

**Table 3 molecules-27-05403-t003:** Texture properties of rice bread added physically modified rice flour.

Sample	Specific Volume (mL/g)	Hardness (g)	Resilence (%)	Cohesion	Springiness (%)	Gumminess	Chewiness
WF	3.72 ± 0.12a	48.59 ± 2.8b	47.18 ± 1.6a	0.89 ± 0.01a	165.74 ± 5.5a	44.15 ± 2.8bc	68.62 ± 4.3b
SF	3.14 ± 0.10b	51.30 ± 4.7b	42.73 ± 1.7b	0.83 ± 0.01b	97.39 ± 1.8b	49.61 ± 5.1b	49.74 ± 6.2bc
HT	1.18 ± 0.07c	701.70 ± 125.5a	33.54 ± 2.1c	0.66 ± 0.02e	81.23 ± 6.2c	468.40 ± 21.3a	442.94 ± 58.9a
MW	3.51 ± 0.19a	46.06 ± 5.2b	30.73 ± 0.9d	0.74 ± 0.02d	93.78 ± 2.7b	38.81 ± 4.7cd	35.91 ± 4.7c
US	3.63 ± 0.08a	42.27 ± 4.0b	32.41 ± 1.4d	0.78 ± 0.02c	101.35 ± 6.6b	32.62 ± 3.4d	31.29 ± 2.8c

Different letters in the same row indicate significant differences (*p* < 0.05). WF: wet-milled rice flour, SF: semidry-milled rice flour, HT: semidry-milled rice flour treated by hydrothermal, US: semidry-milled rice flour treated by ultrasound, MW: semidry-milled rice flour treated by microwave.

## Data Availability

Not applicable.

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
