# Peer review of "Ultrasound Treatment Enhanced Semidry-Milled Rice Flour Properties and Gluten-Free Rice Bread Quality"

_molecules, 2022, doi:10.3390/molecules27175403_

Round 1

Reviewer 1 Report

The manuscript needs only some modifications. It is interesting and innovative, with applicability.

Titles 2.4 and 3.7: this is the name of the technique applied - reformulate the headings to indicate why this technique was applied in the work.

The ultrasound (3.2.1) and microwave (3.2.2) treatment methods should be referenced, as they were not studied by the authors.

Reviewer 2 Report

 Article

Ultrasound treatment enhanced semidry-milled rice flour properties and gluten-free rice bread quality

molecules-1833940

Comments to author

Line no 23: Quality? Nutritional content only or texture and appearance ?

Line no 24: Abstract needs to be approved in terms of novelty character of this research

Line no 36: Physical modification effect on sample properties?

Line no 37: what kind of variations will occur in starch after these modifications? Discuss

Line no 47. Please add these papers regarding ultrasound and potatoes

·         Modeling the drying of ultrasound and glucose pretreated sweet potatoes: The impact on phytochemical and functional groups

·         The quality behavior of ultrasonically extracted sunflower oil and structural computation of potato strips appertaining to deep frying with thermic variations.

Line no 72: You should add a section of material and method here.

Line no 80: Ultrasonic power only effect the starch size of rice flour?

Line no 97: Tables data is appreciable.

Line no 159: do these physical modifications affect protein structure positively?

Line no 204: increase in water absorption effect the shelf life of product?

Line no 254: Graphical representation is very appreciable.

Line no 502: conclusion section should be rewritten in a synthesis and compact manner.

Overall this manuscript is good only some minor changes are required.

Reviewer 3 Report

Qin et al Ultrasound Treatment

L 31 substitute “its” for “it is”

L 36 I would suggest “for environmental and economic considerations”

L 39-41 This sentence is awkward, it needs revision

L 52-56 Extremely long and repetitive sentence. It needs to be rewritten more concisely.

L 64 – were applied not was applied

Fig 1 Page 4 – The treatment ultrasound is missing. The authors used abbreviation that does not match

L 77 US treatment is not found in Fig. 1

Fig. 2 All the micrographs in Panels A appeared to lack sharpness

Table 3 – all the texture values except for cohesion have two and three integers; these are large values and do not need to report the standard division with hundredths (two decimal points). It is useless, two integers reported with tenths SD is more than sufficient, no hundredths please. Anyone with some common and statistics knowledge would agree. I know that dozens of reports are found with this nonsense but that does not make it right.

L 117-118 Honey-like structures are not formed by water removal during freeze drying. Please correct

L121-122 The authors should limit to demonstrate correlations observed – I would like to see documentation of such statement and further substantiation of what they observed and correlations

L128 “physically” modified rice ….

L132-136 It is not clear “similar trend in relative crystallinity and then of Yang et al ---– if it is easier to damage the amorphous region…”  There are several things mentioned and not very clear. Please clarify

L 139-140 starch granules crystal structure or the crystal structure of starch granules nut not “its”

L 152 “slightly reduced” what we need to know first is if it was statistically different and second if that change was small. The way is written is not acceptable.

L 165 erase “obvious”

L 168-170 the authors should emphasize the speculative explanation. Are there cases of similar reports of any type of protein? Otherwise, the results are in clear contrast to expectations. Need depth in this area

Table 1 -OH absorption peak data need statistical analysis, otherwise there is no use of the data. One cannot conclude anything from the data presented and as narrated L 176 -189

L 193-194 significantly lower (insert the p value in parenthesis)

L 202-204 the authors should report associations with the secondary structure data. Otherwise this reads truncated and not relevant

L 341 – and forward:  the proposed groups need to be specifically marked in the graph – this is basic. The reader is left guessing and do all the work when that is the job of the authors. Not  well presented 

L 337 erase “totally”

L 362 “tempered for 35 min with the addition of water.”

L 370 “was prepared as follows”

L 373-374 The sentence makes me wonder if the authors performed the action as stated. To my knowledge the samples are removed when done experientially and that means many things but not as it is written. The samples are analyzed for moisture content later. This is not what is conveyed in the sentence.  Needs to be rewritten for accuracy.

L 479 – the authors have not used the appropriate method for bread texture. “Five replicates” does not mean anything for this test and they are not enough. The minimum number of bread loaves that need to be analyzed per treatment is two. For each bread loaf a minimum of 6 slices need to be analyzed with a total of minimum 12 readings/measurements per treatment. Therefore the values for bread texture are not acceptable.

Round 2

Reviewer 3 Report

The tables and figures 2 and 6 have different sequence of the treatments. Change so all of them will match the sequence of presentation.

L 463  “Each formulation was carried out in duplicate.”    Incomplete description on the baking section. If the authors actually performed an experimental baking experiment one more time after the revision comments were received, how is it that the narrative of that action was not found in the manuscript. The baking reps were done at different dates and analyzed different number of slices per baking date. If the authors actually baked for a third time it is very strange that this was not reflected and reported correctly. reported.

L 318 “from 50.41 g, for the control bread made with SF, up to 701.70 g. The small bread volume”

L 175 In red font are the suggestions to improve the sentence. “There was an increase in α-helix content of US, but the β-turn content of US decreased. These observations may be explained in part by the reorganization of the loose protein structure under cavitation effect of ultrasound”

L 280 correct the subscript 2 in cardo dioxide “strength to bear the rapid expansion of CO2 gas cell during baking. The higher bread vol-“

Please correct throughout the manuscript – X-ray diffraction is an estimation of the overall structure of crystalline or semi-crystalline regions of the samples

L 422 please correctThe FTIR spectra of rice flour samples were obtained using FTIR spectrometer“

L 495 could be considered as an ideal means to is an alternative to improve the morphology and physical proper- 495
